# New Insights into Radio-Resistance Mechanism Revealed by (Phospho)Proteome Analysis of *Deinococcus Radiodurans* after Heavy Ion Irradiation

**DOI:** 10.3390/ijms241914817

**Published:** 2023-10-01

**Authors:** Shihao Liu, Fei Wang, Heye Chen, Zhixiang Yang, Yifan Ning, Cheng Chang, Dong Yang

**Affiliations:** 1State Key Laboratory of Proteomics, Beijing Proteome Research Center, National Center for Protein Sciences (Beijing), Beijing Institute of Lifeomics, Beijing 102206, China; liushihao@ncpsb.org.cn (S.L.); 13472227429@snnu.edu.cn (H.C.); ningyifan@ncpsb.org.cn (Y.N.); changcheng@ncpsb.org.cn (C.C.); 2College of Life Sciences, Hebei University, Baoding 071002, China; 20218017027@stumail.hbu.edu.cn

**Keywords:** *Deinococcus radiodurans*, radiation, proteome, phosphoproteome, dynamic change, kinase

## Abstract

*Deinococcus radiodurans* (*D. radiodurans*) can tolerate various extreme environments including radiation. Protein phosphorylation plays an important role in radiation resistance mechanisms; however, there is currently a lack of systematic research on this topic in *D. radiodurans*. Based on label-free (phospho)proteomics, we explored the dynamic changes of *D. radiodurans* under various doses of heavy ion irradiation and at different time points. In total, 2359 proteins and 1110 high-confidence phosphosites were identified, of which 66% and 23% showed significant changes, respectively, with the majority being upregulated. The upregulated proteins at different states (different doses or time points) were distinct, indicating that the radio-resistance mechanism is dose- and stage-dependent. The protein phosphorylation level has a much higher upregulation than protein abundance, suggesting phosphorylation is more sensitive to irradiation. There were four distinct dynamic changing patterns of phosphorylation, most of which were inconsistent with protein levels. Further analysis revealed that pathways related to RNA metabolism and antioxidation were activated after irradiation, indicating their importance in radiation response. We also screened some key hub phosphoproteins and radiation-responsive kinases for further study. Overall, this study provides a landscape of the radiation-induced dynamic change of protein expression and phosphorylation, which provides a basis for subsequent functional and applied studies.

## 1. Introduction

*D. radiodurans*, a radiation-resistant bacterium, was discovered in 1956 in canned meat after exposure to ionizing radiation. Subsequent studies have shown that the bacteria are highly resistant to a variety of extreme environments including ionizing radiation, such as extreme desiccation, low/high temperatures, oxidative stress, and antibiotics [1,2,3]. When exposed to ionizing radiation, the cell wall structure of bacteria tends to be destroyed [4]. However, the SlpA protein and its unique Hpi protein in *D. radiodurans* form a thick S-layer, enhancing the rigidity of the cell wall and further improving its resistance ability [5,6]. The nucleoid region of *D. radiodurans* exhibits a ring-like structure and can still restore its normal morphology within hours after exposure to 15,000 Gy of ionizing radiation [7]. Different radiation qualities also show different biological effects. In human A549 cells, 2Gy of carbon ions caused the same survival ratio as 6Gy of X-rays [8]. This phenomenon seems to be more pronounced in *D. radiodurans*. In the face of heavy ion radiation, *D. radiodurans* demonstrates the phenomenon of “overkill” [9]. Under 80 Gy of ^12^C^6+^ heavy ion irradiation, the fatality rate of the bacterium reaches nearly half, then decreases with increasing doses, reaching a minimum at 160 Gy, and subsequently recovers. For comparison, the fatality rate of *E. coli* was up to 95% at 5 Gy and continued to be at a high level when the was dose increased [10].

Previous studies have discovered that *D. radiodurans* possesses DNA repair proteins which are similar to those found in most other prokaryotic organisms, including proteins in base excision repair (Ung, Mug, AlkA2, MutM, NTH, etc.), double strand break repair (UvrA, RecA, RecD2, RecO, RecR, RecF, RecQ, etc*.)*, mismatch repair (MutS, MutL, UvrD, XseA, RecJ, SSB, DpoⅢ, Lig, etc.), nucleotide excision repair (UvrA, UvrB, UvrC, UvrD, Mfd, DpoⅠ, etc.) [11,12,13,14,15]. But, compared to radiation-sensitive species, these proteins in *D. radiodurans* may play some more effective roles in DNA repair. Deinococci-specific proteins, such as PprA, DdrA, DdrB, DdrC, DdrD, have also been confirmed to be highly expressed when exposed to irradiation, helping to maintain genome integrity [16,17,18]. Besides that, *D. radiodurans* also encodes two proteins named ‘DNA protection during starvation protein’ (Dps-1, Dps-2); both can store iron and manganese in ionic states and protect DNA from oxidative damage [19,20,21]. For the oxygen radicals that have already formed, the catalases in *D. radiodurans* and the abundant H-Mn^2+^ complex play a major role in their removal [1,22].

Post-translational modifications cannot only affect the structure, activity, stability, and cellular localization of a single protein, but also regulate the function and activity of a series of proteins. Succinylation proteomics studies revealed the regulatory role of lysine succinylation in *D. radiodurans*, indicating the important role of PTMs in its extreme resistance [23]. Protein phosphorylation (mainly on serine/threonine/tyrosine), as one of the most common and essential PTMs in eukaryotes, can rapidly regulate protein activity and achieve rapid adaptation to the environment [24,25]. Phosphorylation may also be widespread in bacteria and affect a variety of life activities, including radiation response. RqkA, a DNA damage responsive serine/threonine protein kinase, plays a pivotal role in DNA repair and cell cycle in *D. radiodurans* [26,27]. The phosphorylation of Tyr-77 and Thr-318 had been shown to regulate the activity of DrRecA, thereby increasing its affinity for dsDNA [28]. FtsZ protein is involved in bacterial cell division, and the phosphorylation of DrFtsZ at S235 and S335 causes the loss of its function and reduces the cell tolerance to the gamma ray [29]. However, a smaller number of phosphosites have been identified in bacteria compared to eukaryotes. Even in *Escherichia coli*, a common model organism, fewer than 1600 high-confidence class I phosphosites (localization probability ≥ 0.75) have been identified [30]. Only a few of them have functional annotations [31]. The profiling and in-depth functional analysis of the dynamic phosphoproteome of *D. radiodurans* will help us to understand the response mechanisms to high doses of radiation.

In this study, we used label-free proteomic technology to detect the dynamic changes of the (phospho)proteome of *D. radiodurans* under different heavy ion doses and different time points after irradiation. The dynamic changing patterns and functional interpretations were explored. The pathways and the key hub phosphoproteins, as well as kinases related to irradiation resistance were screened for further studies.

## 2. Results

### 2.1. Global Analysis for (Phospho)Proteome of D. radiodurans after Heavy Ion Irradiation

In order to detect the dynamic changes of protein levels and phosphorylation modification levels after *D. radiodurans* irradiation, we selected three radiation doses (20 Gy, 80 Gy, 160 Gy) [10] and two time points after irradiation (0h, 2h), with a control group at each time point. High-resolution LC-MS/MS was used for peptide identification and relatively accurate protein quantification was obtained (Figure 1A). We quantified 2359 proteins, more than all published data to date [10,32,33] (Figure 1B), with more than 2000 proteins identified per sample on average (Figure 1C, Appendix A). In addition, we identified 1110 class I phosphosites among 568 phosphoproteins (Appendix A), including 587 (53%) serine sites and 407 (37%) threonine sites (Figure 1C). Almost all phosphoproteins have been observed and quantified in the proteome (Appendix A). The violin plot shows the quantification, median, quartile, and expression distribution of each experiment, with no significant difference in each of the 16 measurements of the proteome and phosphoproteome, demonstrating the stability of our dataset (Appendix A). Using PCA (principal component analysis) to assess the variability between different samples of the proteome, we found that samples of the same type cluster together, while samples at different time points or irradiated with different doses could be separated clearly, indicating the repeatable variation between different sample groups (Figure 1D). Spearman correlation analysis found that the correlations between the two repeats were 0.89–0.92, further indicating that the experiment had good repeatability (Appendix A).

Of all the proteins identified, 1553 (~66%) were significantly changed after irradiation (two-way ANOVA, FDR < 0.05, Figure 1E, Appendix A). The counts of upregulated proteins (FDR < 0.05, log_2_FC > 1) were generally higher than those of downregulated ones (FDR < 0.05, log_2_FC < 1), and this trend was more obvious with the increase in radiation doses and time points (Figure 2A, Appendix A). A total of 940 proteins were upregulated in at least one radiation state. Compared with three previous irradiation datasets of *D. radiodurans*, 33 upregulated proteins were identified in at least three datasets, 32 of which were identified in our data (Appendix A). In contrast, the phosphorylation level of 255 (~23%) sites significantly changed after irradiation, less than the proportion of differential proteins (Figure 1E, Appendix A). However, we found an interesting phenomenon that the upregulation of the phosphorylation level (median log_2_FC = 5.22) was much higher than the level of variation in protein abundance (median log_2_FC = 1.23) (Figure 1F). This means that a smaller number of differential phosphosites are sensitive response sites for irradiation in *D. radiodurans*. Their functional role in radiation resistance needs further in-depth analysis.

### 2.2. Dose and Stage Dependence of D. radiodurans Irradiation Response

We conducted a hierarchical clustering analysis of the 1553 differently expressed proteins (DEPs) after being exposed to irradiation (Figure 2B). The analysis revealed that for samples collected immediately after radiation (0h), there was a clear distinction between DEPs at different doses. However, in samples collected 2h after irradiation, this distinction was not apparent (Figure 2B). The number of upregulated proteins was generally higher than that of the downregulated ones, and this trend was more obvious with an increase in doses (Figure 2B, Appendix A). Statistics on the number of upregulated proteins at different doses also showed that the samples at 0h after irradiation were significantly differential at each dose, and 178 (23.5%) of the three doses were upregulated together (Figure 2C). Meanwhile, the number of co-upregulated proteins reached 240 (42%) in the samples collected 2h after irradiation (Figure 2D). These results reflect the characteristics of the early and late radiation responses of *D. radiodurans*.

Since the 0h samples have their own characteristics at each dose, function enrichment analysis was performed for the proteins upregulated at each state alone and the proteins upregulated in all doses. Gene ontology (GO) analysis showed that 20 Gy-specific upregulated proteins tend to participate in the cytokinin biosynthetic process (e.g., DR_0471) and trehalose biosynthetic process (e.g., DR_0464, TreZ). TreZ is a key protein in trehalose synthesis in bacteria [34], and trehalose acts as a protector against various stress conditions [35]. In addition, heat shock protein binding entries (e.g., DnaJ) were also enriched. With the increase in irradiation dose, the expressions of proteins related to transmembrane transport and tetrapyrrole metabolism were upregulated. Proteins related to ligase activity were upregulated at various radiation doses, including the proteins related to the ligation between tRNA and amino acids (e.g., GlyQS, PurD, GatC, ProS), and tRNA methylation (e.g., RtcB) [36,37,38,39]. These processes are related to accurate translation, which is important for cell repair after irradiation.

From the viewpoint of different stages after irradiation, the upregulated proteins at different time points showed obvious different functional characteristics. Proteins related to ATP-dependent DNA metabolism and phosphorylation were immediately upregulated after irradiation, which suggests the significance of phosphorylation for the *D. radiodurans* radiation response. It is noteworthy that membrane proteins and transmembrane transport proteins were upregulated during irradiation recovery (2h). Receptors on the cell membrane can sense external radiation stimuli and initiate a series of intracellular reactions through signal transduction pathways [40]. Transporters in the membrane help to maintain cell homeostasis and are involved in maintaining membrane integrity [41]. For example, LrgA and LrgB can form protein complexes that exert anti-apoptotic functions by maintaining the integrity of cell membranes [42]. Proteins related to signal transduction and nucleic metabolism were upregulated immediately after irradiation and remained in an upregulated state after 2h. Specifically, sustained upregulated proteins were enriched in the positive regulation of signal transduction, such as TOR signaling and nucleotide biosynthetic process. Protein domain enrichment analysis showed similar results to GO functional analysis (Appendix A).

### 2.3. KEGG Pathway Analysis of Differentially Expressed (Phospho)Proteins

The pathway data of *D. radiodurans* were downloaded from the KEGG website (https://www.kegg.jp) [43] and matched with the (phospho)proteome (Appendix A). Most of the proteins (91.7%) in the pathways were quantitatively identified. Several proteins in the RNA polymerase (e.g., RpoC, RpoZ) and RNA degradation (e.g., Eno, Rny, GroEL) were upregulated, as well as the level of phosphorylation modification (Table 1). In the antioxidant pathways, such as alpha-linolenic acid metabolism, phenylalanine metabolism, and riboflavin metabolism, the levels of protein expression and phosphorylation were all upregulated to different extents (Figure 3A). These pathways may help to clear radiation-generated ROS and maintain cellular homeostasis [44,45,46]. Notably, 20 of the 29 identified proteins in the oxidative phosphorylation pathway were upregulated, including multiple subunits of NADH dehydrogenase and V-type ATP synthase, and cytochrome c oxidase (Appendix A). ATP produced by the mitochondrial respiratory chain is necessary for maintaining the energy supply of cells and for normal physiological functions and metabolic activities. The expressions of many proteins in the pathways of homologous recombination (RecFOR pathway) and mismatch repair were upregulated in at least one state after irradiation (Figure 3B). Due to the lack of RecB/C/D protein, the RecFOR pathway plays a key role in the recombination repair of *D. radiodurans* [47]. The key proteins RecF/R in this pathway were upregulated in all radiation states, with the highest fold change up to more than 35. The fold change of RecO in ‘160Gy, 0h’ also reached about 5. The activation of the core proteins in the mismatch repair pathway, MutS and MutL, helps chromosomes to reshape quickly to maintain genome stability.

### 2.4. Dynamic Patterns of Phosphorylation and Functional Interpretation

Preceding analyses revealed that the upregulation of phosphorylation is significantly higher than the upregulation of protein level, suggesting that phosphorylation may play an important role in the irradiation response. In order to better understand the distribution of 255 differential phosphosites, Fuzzy c-means clustering was used to divide them into four clusters, which showed distinct differences (Figure 4A). Cluster 1 (n = 56) phosphosites were upregulated in the early stage and recovered in the late stage, cluster 2 (n = 66) phosphosites were not significantly changed in the early stage and upregulated in the late stage, and cluster 3 (n = 68) phosphosites showed continuous upregulation, while cluster 4 (n = 65) phosphosites showed downregulation. Motif analysis showed that RS/RT motifs were enriched in different expression phosphosites (Appendix A). For these significantly changed phosphoproteins, we downloaded the protein–protein interaction (PPI) data from the STRING database and constructed a PPI network [48]. A total of 25 core regulatory proteins were screened (Figure 4B). DR_0183, with the highest score, has the glutamine amido transferases class II domain, participating in GABA biosynthesis, which has been reported to develop a significant response to radiation stress in plants [49,50,51]. In addition, a series of proteins, including RpoB, a core enzyme for RNA synthesis, have phosphosites with different dynamic patterns, which may be regulated by different kinases and play different parts in their functional activities.

According to their functions, differential phosphoproteins are mainly divided into nine classes, involved in different vital activities (Figure 4C). There are multiple metabolic proteins, which participate in glycometabolism, lipid metabolism, and the metabolism of other nutrients, and these account for a considerable proportion. In the proteins located in the cell wall, S460 in Deinococcus-specific HPI protein showed late upregulation. Another protein SlpA was upregulated late at T243, and the other three sites were downregulated (Y170, S241, T661). The phosphorylation of ribosomal proteins (RpsN, RpsL, RplB, RplY) and heat shock proteins (GroEL, GroES, DnaK) also showed upregulation. Previous studies have suggested that there is also a relationship between protein disorder and stress response [52]. Thus, we calculated the disorder degree for each amino acid residue and classified the proteins into different disorder categories (Appendix A). FtsKL and DR-specific protein DdrD were identified as disordered proteins, and their upregulation sites (S905 in FtsKL, cluster 2; S170 in DdrD, cluster 3) are located in disordered regions. A further analysis revealed that the phosphosites identified by *D. radiodurans* tended to be located in disordered regions (Appendix A), similar to the situation in eukaryotes [53].

### 2.5. Comparative Analysis of Phosphosites and Protein Expression

To determine whether changes in phosphorylation are due to the changes in protein levels, we analyzed the consistency of the two omics using log_2_FC standard (|log_2_FC| ≤ 1 means no change). The proteins were classified using “both up”, “both down”, “both no change”, “pro > pho”, and “pho > pro” (Figure 5A). Each radiation state was calculated separately (Figure 5B). If one protein has multiple phosphosites, and the dynamic pattern of these sites and the protein level meet different classification criteria, the protein will be counted multiple times. We found that the two omics were inconsistent in the majority of cases, i.e., more than 50% in each irradiation state. The class of “both up” increased after 2 h irradiation, but still maintained a lower level. This suggests that radiation stress specifically regulates proteins primarily at a single omics level.

Functional analysis was performed for proteins inconsistent in the two omics levels and for proteins that were both upregulated in the two omics (Figure 5C). The “pro > pho” class was enriched in lipid metabolism and protein folding. The “pho > pro” class proteins were related to nucleic acid metabolism, translation, and the cellular response to an abiotic stimulus. These terms are more closely related to stress response and repair. For example, we were surprised to find that the protein level of RpoB did not change significantly in any state, but there were significant changes in its six phosphosites (Figure 5D). This suggests that phosphorylation is essential for its function after irradiation.

### 2.6. Irradiation-Sensitive Proteins Containing Kinase Domains

Studies on the kinome of *D. radiodurans* are lacking at present. In this study, we identified 79 proteins containing kinase domains, 52 of which were sensitive to irradiation, including 35 proteins that were upregulated in at least one state (Figure 6, Appendix A). In the annotation of its domains and functions, we found 20 protein kinases, 11 nucleotide kinases, and four lipid kinases. An analysis of the homologous proteins of these kinases revealed that DR_A0332 (a non-specific serine/threonine protein kinase) is a protein kinase only present in bacteria. We also identified six site-specific kinases; the phosphorylation modification levels of these kinases were upregulated, while protein levels did not change significantly. This also included three glucokinases (DR_2635, PckA, GlgC), two lipid kinases (GGlpK, DR_1363), and a nucleotide kinase (DR_A0020).

## 3. Discussion

In this study, high-precision mass spectrometry and label-free quantification techniques were used to study the dynamic changes in the protein and phosphorylation level in *D. radiodurans* after heavy ion irradiation. In total, we quantified more than 2300 proteins and 1100 high-confidence phosphosites, which is the first phosphorylation dataset and the largest proteome dataset of *D. radiodurans* to date. Compared with previous studies, our criteria were more stringent and the results were better (Appendix A). In total, 74.9% of the upregulated proteins were unique in our dataset (Appendix A) [10,32,33]. Based on the integrated analysis of this high-quality dynamic (phospho)proteome dataset, several new insights into radio-resistance mechanisms were observed.

First of all, compared with protein abundance, the phosphorylation level has a much more significant upregulation after irradiation (Figure 1F). This phenomenon suggests that protein phosphorylation is more sensitive to irradiation than the expression level of proteins. A previous study also found this trend in the CVB3-virus-infected HeLa cells [54], but this is the first time that irradiated samples are being analyzed. Notably, the difference in the upregulation degree between the protein expression level and phosphorylation is much higher in *D. radiodurans* of this study. Thus, deciphering the dynamic changes in phosphorylation after irradiation is of great significance to fully understand the mechanism of extreme ability of radio tolerance in *D. radiodurans.*

Second, the dynamic change in the (phospho)proteome at different time points after irradiation could provide valuable information to understand the mechanisms of the immediate response and further repair process. However, prior to this study, there were no studies focusing on this question. Through the identification and in-depth analysis of samples at different time points after irradiation, we found that the radiation response of *D. radiodurans* is time-dependent. The functions of upregulated proteins at different time points were significantly different (Figure 2F). Early upregulated proteins were involved in stress-related functions, while late upregulated proteins were enriched in repair-related functions. For the significantly changed phosphoproteins, we distinguished four distinct expression patterns according to the dynamic changes at different time points (Figure 4A). Further detailed functional studies will provide more specific clues for the mechanism of irradiation response and repair.

Third, the significantly upregulated (phospho)proteins identified in this study could serve as potential radiation-responsive key molecules for further functional research. In the list of these proteins, a variety of known radiation-responsive proteins, including IrrE, PprA, DdrA, DdrD, PolA, PolB, UvrB, UvrC, RecF, RecO, RecR, are involved, proving the reliability of our research strategy. For the new potential key proteins identified, we further analyzed the possible interaction networks they may form and their potential functions. For example, a PPI network was constructed to screen core phosphoproteins. Three phosphoproteins, DR_0183 (large subunit of glutamate synthase), DR_0269 (a DR-specific highly disordered protein), and Eno (Enolase), were the top ones with the highest connectivity.

Fourth, we found that more than half of the radiation-responsive proteins showed inconsistent changes in protein expression levels and phosphorylation levels (Figure 5A,B). Among them, RpoB, a key protein in the RNA synthesis pathway, has no significant change in its protein expression level, but its expression is upregulated after irradiation at multiple phosphosites. RpoB protein can bind to rifampicin to inhibit transcriptional initiation. Previous studies evaluated the mutation of this gene by analyzing rifampicin-resistant clones to estimate the extent of DNA damage caused by radiation from the space station [55], but the link between its phosphorylation and radiation resistance was lacking in previous studies.

Finally, the results of this study may have potential application value. Some radiation-activated pathways, such as the mismatch repair pathway, RNA metabolic pathway, and antioxidant pathways (alpha-linolenic acid metabolism, phenylalanine metabolism, riboflavin metabolism), also exist in humans. As an essential fatty acid, alpha-linolenic acid has anti-inflammatory, anticancer and anti-metabolic syndrome properties [56]. Riboflavin, a type of vitamin B2, can play a protective role in nervous system diseases by regulating the mitochondria energy production–oxidative stress pathway [57]. Our study provides support for the potential anti-radiation applications of these edible organic compounds.

In summary, this study comprehensively analyzed the (phospho)proteome dynamic changes induced by heavy ion radiation from the aspects of differential expression, function, pathway, protein–protein interaction, subcellular localization, etc., which provide a fundamental framework for a system understanding of this radiation-tolerant organism. In particular, our discovery of core phosphoproteins, radiation-responsive kinases, and multiple radiation-responsive pathways will help us to better understand the underlying mechanisms and to build a foundation for further applied research.

## 4. Materials and Methods

### 4.1. Strain and Culture Conditions

*D. radiodurans* strains were purchased from the China General Microbiological Culture Collection Center (No. 1.3828, CGMCC, Beijing, China). The strains were cultured in TGY liquid medium (1% tryptone, 0.5% glucose, and 0.1% yeast extract) at 30 °C with shaking at 200 r/min.

### 4.2. Pre-Irradiation Preparation, Irradiation, and Post-Irradiation Treatment

*D. radiodurans* strains were cultivated to the early stationary phase (OD600 = 1.5). Samples were collected at 4000 r/min, at 4 ℃ for 3 min, and were washed three times with 1× PBS, and then finally re-suspended with 1× PBS (400 μL). The samples were subjected to heavy ion irradiation doses of 20 Gy, 80 Gy, and 160 Gy, respectively, at room temperature with a dose rate of 80 Gy/min (HeavyIon Research Facility in Lanzhou (HIRFL), Institute of Modern Physics, Chinese Academy of Sciences). In addition, a control group without exposure was set up. Immediately after irradiation, 1/2 of the samples were collected at 13,300 r/min for 5 min, and stored at −80 ℃. The remaining 1/2 samples were left at room temperature, and were collected after irradiation for 2 h.

### 4.3. Protein Lysis and Digestion for (Phospho)Proteomics Analysis

A total of 8M UA solution containing 1× protease and phosphatase inhibitor (Thermo, USA) was added to the bacterial precipitation for cell lysis and placed on ice for 30 min. Then, it was crushed on the ice with ultrasonic (SCIENTZ, China), and centrifuged at 4 ℃ at 14,000 r/min (Thermo, USA) for 5 min. Then, the precipitation and the surface oil were discarded, the supernatant was taken, and the protein concentration was determined using the BCA kit (CWBIO, China). In total, 1.5 mg protein was added to DTT (final concentration was 10 mM), bathed in water at 56 ℃ for 30 min, lowered to room temperature, added into 1 mol/L IAA solution, shielded from light for 30 min, and centrifuged at 14,000 r/min for 10 min. The protein supernatant was transferred to two 10 kD ultrafiltration tubes (Millipore, USA) for loading, and centrifuged for 14,000 r/min for 15 min. Then, 8M urea solution was added, followed by vortex oscillation, centrifuging at 14,000 r/min for 15 min 3 times, and the waste liquid was discarded. Next, 50 mM ammonium bicarbonate was added, vortexed for 5~10 s, centrifuged for 15 min 3 times at 14,000 r/min, and then the waste liquid was discarded. The new casing was replaced, 50 mM ammonium bicarbonate and 1 μg/μL trypsin (mass ratio 1/50, Promega, China) were added, shaken, and mixed, and finally digested at 37 ℃ for 12 h. This was centrifuged at 14,000 r/min for 10 min and the peptide solution was collected. Then, 1/10 of the peptide solution was taken in a new centrifuge tube for whole-protein sRP-HPLC identification, the remaining liquid was acidified with 0.1% formic acid, and then all peptide segments were frozen dry (Eppendorf, German) and stored at −80 ℃.

The lyophilized peptide was dissolved in a solution containing 1M lactic acid, 70% acetonitrile, and 5% TFA (Sigma, USA). The peptide was adsorbed on TiO_2_ (GL science, Japan) on a suspension apparatus. The peptide was washed with 30% acetonitrile, 0.5% TFA solution, 80% acetonitrile, and 0.5% TFA solution successively 3 times. The 2%, 5%, 8%, 10%, and 40% acetonitrile solutions were gradient-eluted once each, combined into 3 fractions, acidified, desalted, and identified via high-precision mass spectrometry (Thermo Scientific ™ Orbitrap Exploris 480) [58].

### 4.4. Chromatography and Mass Spectrometry

The 500 ng peptides were loaded onto 20 cm columns packed in-house with C18 particles (1.9 μm). The poly-peptide samples were dissolved in 10 μL of solvent A (0.1% FA in water), injected with 5μL, loaded onto the pre-column at a flow rate of 3 μL/min on the EASY-nano-LC chromatography system, and then separated on the column at a flow rate of 300 nL/min. The gradient was as follows: solvent B (0.1% FA in 80% ACN) increased linearly from 7% to 12% (0 min–10 min); solvent B rose linearly from 12% to 45% (10 min–70 min); and then solvent B was at 95% (55 min–90 min). The mass spectrum data were collected using FAIMS Pro, equipped with specific parameters that were set as follows: The spray voltage of ion source was set to 2.1 kV. FAIMS Pro adopts two voltages of −45 V and −65 V, and each cycle time was set to 1.3 s. Data-dependent acquisition with the MS method was used in one full scan (350–1500 m/z, R = 60,000 at 120 m/z, maximum injection time 50 ms), first with a target of 3 × 10^6^ ions, followed by data-dependent MS/MS scans with higher-energy collision dissociation (AGC target 7.5e^4^ ions, maximum injection time at 22 ms, isolation window 1.6 m/z, normalized collision energy 27%, and R = 15,000 at 120 m/z). A dynamic exclusion of 20 s was enabled.

### 4.5. Database Searching, Quality Control and Data Normalization

The (phospho)proteome data were searched using Maxquant (version 2.0.3.0). The “match between runs” parameter was selected. The iBAQ (proteome) and intensity (phosphoproteome) data were normalized via median normalization. For the calculation of FDR values, we used the R function “anova” for processing normalized data. The Prcomp function was used for principal component analysis. Missing values were interpolated with global minimum values if necessary.

### 4.6. GO and KEGG Annotation

The GO annotation of *D. radiodurans* was downloaded from the UniProt database, in which there are 2103 annotated genes, and we traced the results, which finally corresponded to a total of 3186 GO terms. Pathway information was downloaded from the KEGG website and matched with the UniProt accession. Annotation information for all genes is listed in Appendix A.

### 4.7. PPI Network Analysis

The STRING database was used to download the interaction network of all differential phosphoproteins. We selected the default parameters in the basic setting. Then, we drew the network diagram using Cytoscape [59]. Since some proteins have multiple differential phosphosites, we used different colors to indicate the cluster that the sites belonged to.

### 4.8. Protein Structure and Evolutionary Age Annotation

Protein domains were identified using the HMMER-scan program to search the Pfam databases. All Pfam-A domain entries were used to identify domains in each protein [60]. Finally, we obtained 2490 proteins with domain annotations. The domain information for all proteins is shown in Appendix A.

Intrinsically disordered regions were identified using SPOTD (default parameters). SPOTD is used to calculate the structural disorder value of each amino acid residue [61,62]. An amino acid residue with a high structural disorder value, which exceeds the default threshold (0.426), was considered to be disordered. After we obtained the results, the structural disorder ratio (SDR, i.e., the percentage of disordered residues) was calculated and classified into 3 grades: 0–10%, highly structured; 10–30%, moderately disordered; 30–100%, highly disordered [63]. Then, we matched the result with phosphorylation site to determine whether the site was disordered or not.

To determine the evolutionary age of each gene in *D. radiodurans*, a phylostratigraphy approach [64,65,66] was used. We selected representative species with complete genome sequences in the UniProt database. In each genus, the species with the highest number of protein sequences was selected. In addition, all species of the Deinococcus genus were selected to determine whether the gene was genus-specific or species-specific. In total, 3394 species remained. We performed reciprocal alignment of all *D. radiodurans* protein sequences with the protein sequence of the other species using Diamond BLASTP [67]. The matches with both e-values less than 1E-5 were regarded as homologous genes. At last, we determined the gene age of each gene, and classified them into 4 grades: Cellular organism, Bacteria, Deinococcota, and *Deinococcus radiodurans*. Finally, we identified 498 Deinococcota-specific proteins and 164 *Deinococcus radiodurans*-specific proteins.

## Figures and Tables

**Figure 1 ijms-24-14817-f001:**
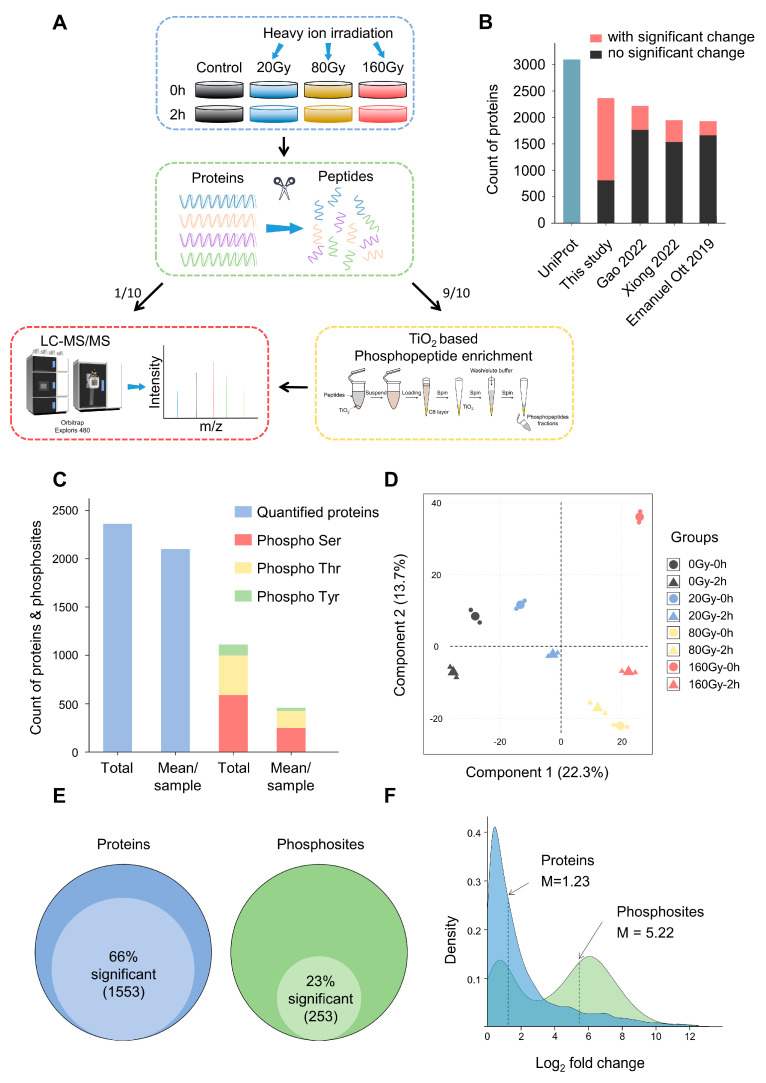
Experimental design, quality control and overview of (phospho)proteome data. (**A**) Experimental design for (phospho)proteome analysis of heavy ion irradiated *D. radiodurans*. Strains were treated with 3 doses of radiation from 20 to 160 Gy. Samples were collected at two time points (0 h, 2 h) after irradiation. The color scheme for each dose remained essentially the same throughout the paper. (**B**) A comparison with the count of proteins identified by *D. radiodurans* proteomics published in the last 3 years [10,32,33]. (**C**) The number of proteins and phosphosites identified as a whole and the average number identified per sample. (**D**) PCA analysis of the global proteome samples. Different colors/shapes represent different doses/time points. (**E**) Fraction of proteins and phosphosites dynamically regulated after multiple dose irradiation, determined by using a two-way ANOVA test (FDR < 0.05). (**F**) Distribution of the scale of changes of significantly regulated proteins and phosphosites, showing that the differences in phosphorylation are generally more extensive than those at protein levels.

**Figure 2 ijms-24-14817-f002:**
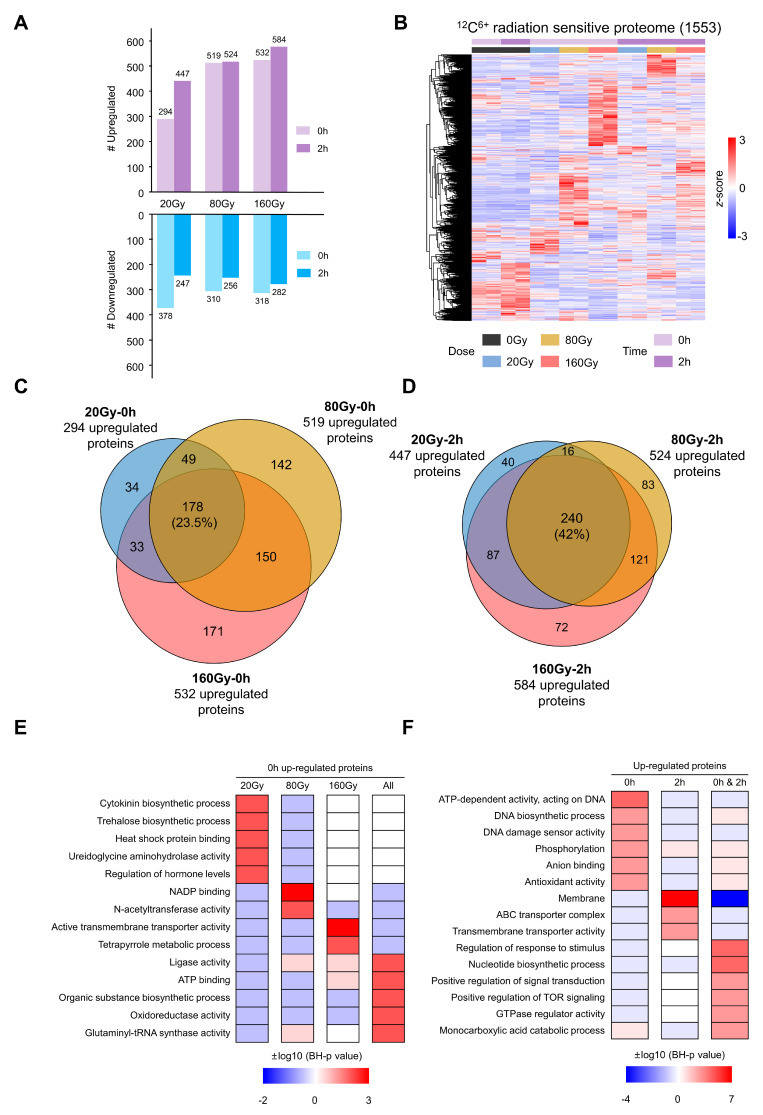
Integration analysis of irradiation sensitive proteins. (**A**) The number of up- or downregulated proteins at different irradiation doses and time points. (**B**) Heatmap of z-scored protein abundance (iBAQ value) of the DEPs. (**C**,**D**) Venn diagram of the upregulated proteins at 3 radiation doses, and 0h (**C**) or 2h (**D**) after irradiation. (**E**,**F**) Function enrichment analysis of 0h upregulated proteins with different doses (**E**), and of upregulated proteins at different time points (**F**). The red scale represents over-representation, while the blue scale represents under-representation.

**Figure 3 ijms-24-14817-f003:**
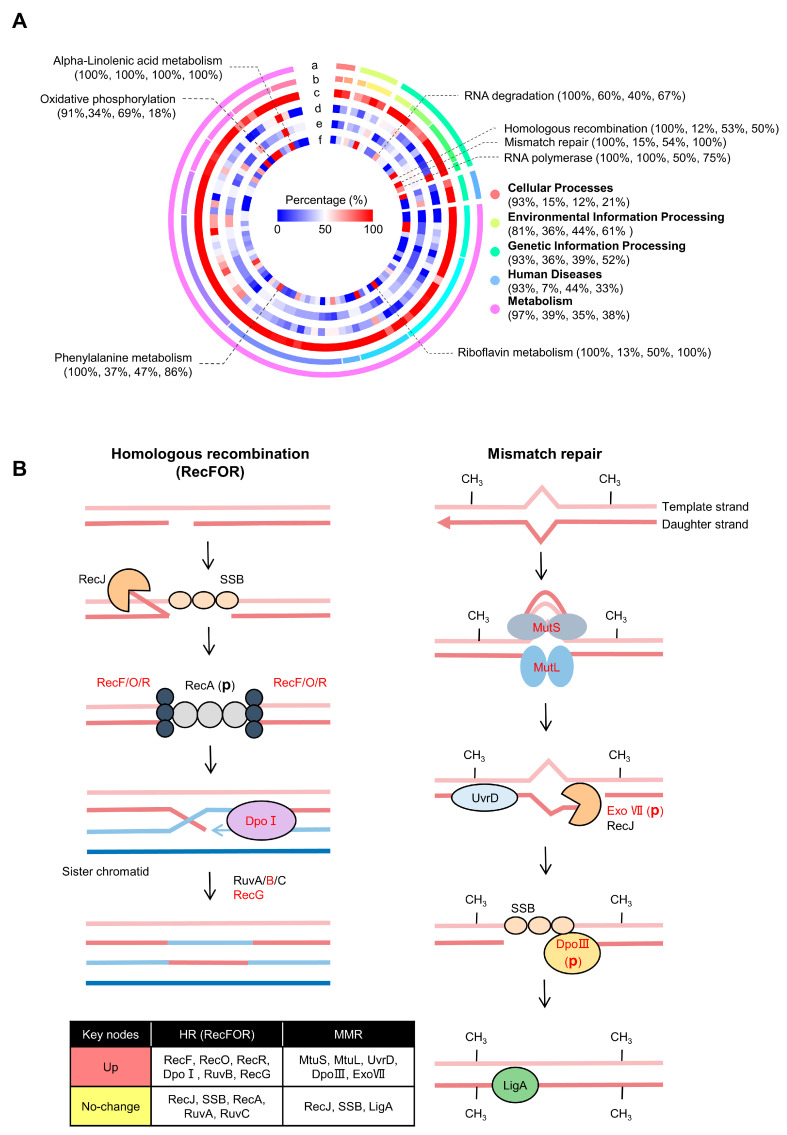
Irradiation response-related pathways. (**A**) The changes in all pathways of *D. radiodurans* after irradiation. The ring diagram from outside to inside is as follows: pathway class (a), pathway subclass (b), proportion of proteins identified (c), proportion of upregulated proteins (d), proportion of phosphoproteins (e), and proportion of proteins with upregulated phosphorylation level (f) (Appendix A). (**B**) Homologous recombination (RecFOR) and mismatch repair pathways. Red color indicates upregulation; (p) means phosphorylation in the protein. The differential expressions of proteins in the pathways are labeled in the table below.

**Figure 4 ijms-24-14817-f004:**
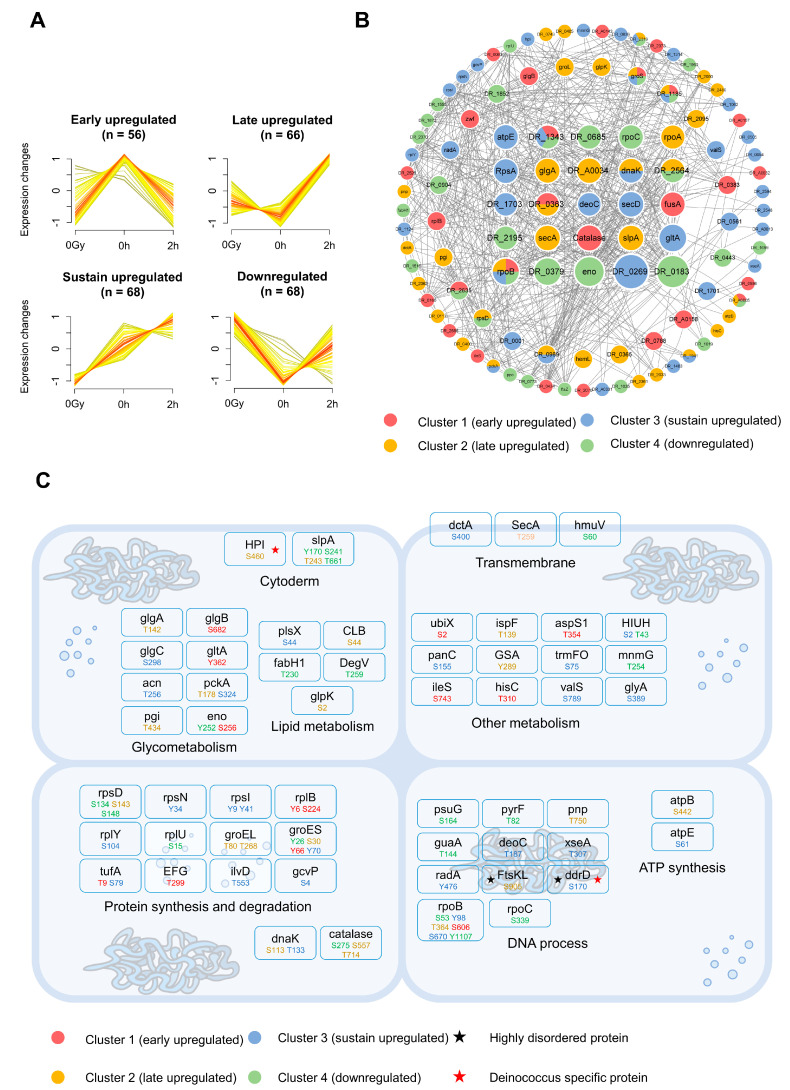
Integration analysis of phosphoproteins and phosphosites. (**A**) Fuzzy c-means clustering of 4 dynamic patterns of differential phosphosites. (**B**) PPI network of interactions between proteins with differential phosphosites. Displayed network includes 25 core proteins and 85 suggested proteins that have at least one interaction with other protein(s). (**C**) Overview of major functional classification of proteins with differential phosphosites.

**Figure 5 ijms-24-14817-f005:**
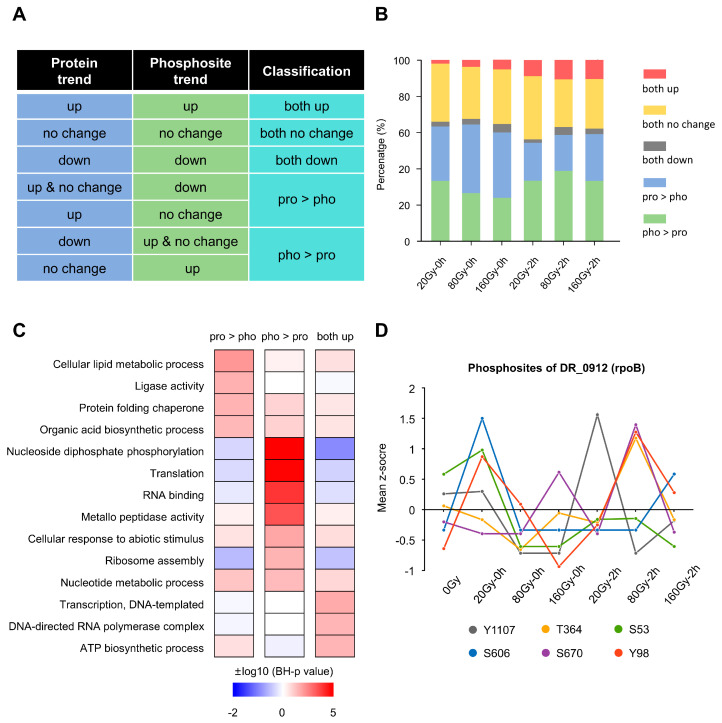
Comparative analysis of protein expression and phosphorylation modification. (**A**) Classification of protein expression and phosphorylation levels after irradiation. (**B**) The proportion of different classes in different irradiation states. (**C**) Functional enrichment analysis of various classified proteins. (**D**) Dynamic changes in modification levels of site-specific protein RpoB.

**Figure 6 ijms-24-14817-f006:**
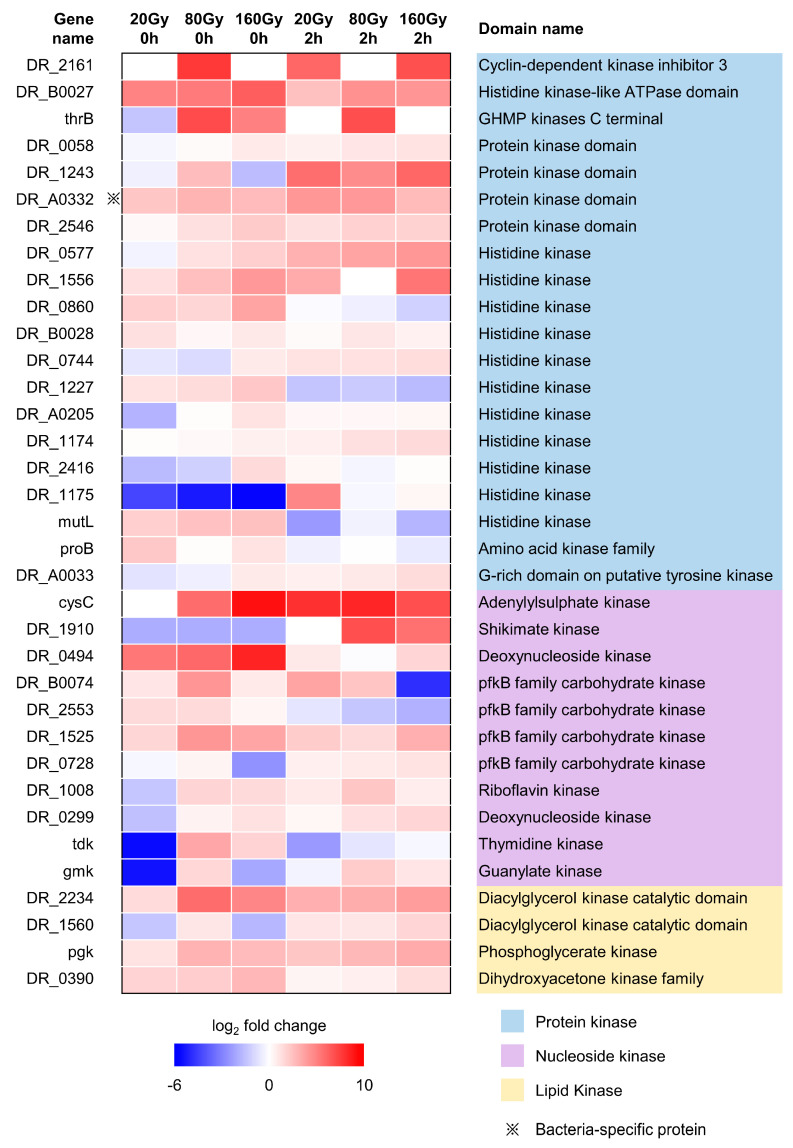
The classification of proteins upregulated in at least one state after irradiation which have a kinase domain. Blue to red color gradient denotes downregulation to upregulation compared with unirradiated samples at this time point. Fold changes were log2 transformed. The text and color on the right, respectively, indicate the name and category of the kinase domain corresponding to the protein. The protein labeled with ※ is a bacteria-specific protein.

**Table 1 ijms-24-14817-t001:** RNA metabolism-related upregulated proteins or differential phosphosites.

Uniprot ID	Protein Name	The Upregulated Experiment Groups Dose (Gy)-Time (h)	Phosphosites Amino Acid-Cluster
Q9RV39	A-adding tRNA nucleotidyltransferase	160-0	NULL
Q9RR60	Enolase	20-0, 80-0, 160-0,20-2, 80-2, 160-2	T252-4, Y256-1
Q9RSR1	Polyribonucleotide nucleotidyltransferase	NULL	T750-2
Q9RY23	Chaperone protein DnaK	20-0, 80-0, 160-0,20-2, 80-2, 160-2	S113-2, T133-3
Q9RWQ9	Chaperonin GroEL	20-0, 80-0, 160-0,20-2, 80-2, 160-2	T80-2, T268-2
Q9RSJ6	RNA polymerase subunit alpha	NULL	Y28-2
Q9RVV9	RNA polymerase subunit beta	NULL	S53-4, Y98-3, T364-2, S606-1, S670-3, Y1107-4
Q9RVW0	RNA polymerase subunit beta’	80-0, 160-0, 20-2, 160-2	S339-4
Q9RRJ6	RNA polymerase subunit omega	80-0, 160-0, 160-2	NULL

## Data Availability

All data generated or analyzed in this study are included in this published article and the Appendix A. Appendix A are available at IJMS online.

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
