# Peer review of "New Insights into Radio-Resistance Mechanism Revealed by (Phospho)Proteome Analysis of Deinococcus Radiodurans after Heavy Ion Irradiation"

_ijms, 2023, doi:10.3390/ijms241914817_

Round 1
Reviewer 1 Report
This manuscript presents a proteomics study of the radiation-induced changes in the proteome and phospho-proteome of Deinococcus radiodurans after exposure to heavy ion radiation. To my knowledge, this work constitutes the first evaluation of the phosphoproteome of this radiation resistant bacterium, and as such represents an important contribution to the field. However, the analysis of the results is very limited and the authors do not discuss their findings in light of earlier proteomics studies. Moreover, the proteomics part of this work appears to be largely a repeat of previously published work (Gao et al, BMC Microb, 2022) and yet there is barely any comparison with this very similar study. Several important issues should thus be addressed (see below) before this manuscript can be considered for publication.
1) The introduction contains many errors regarding notably the DNA repair machinery and the Dps proteins. Important references to earlier work on these targets in D. radiodurans are missing. The lists of DNA repair proteins for each pathway are incomplete and NER is completely missing. Protein names should have a capital letter at their start. ‘Dryness’ should be replaced by ‘desiccation’. The ‘overkill’ phenomenon should be explained as well as the rationale for using these low doses, as compared to numerous earlier studies on D. radiodurans that make use of gamma- or UV-irradiation at high doses (>1000 Gy). There have been several studies on protein phosphorylation in D. radiodurans, and in other Deinococcus species – these should be cited and described – what is known so far about the role of phosphorylation and PTMs in general in D. radiodurans.
2) Can the authors comment on the fact that their study reveals a much higher proportion of proteins exhibiting significant change in expression after irradiation (Fig. 1B) compared to earlier studies and notably to the work by Gao and colleagues performed under very similar conditions? More globally, the proteomics analysis should be compared to the work by Gao and colleagues and should also be compared to earlier proteomics studies – are the same proteins upregulated and downregulated? This needs to be extended in the discussion.
3) In the Mat and Meth, it is unclear whether the cells were returned to the incubator to recover for 2h after irradiation. Were the cells transferred to fresh medium? Was the irradiation performed at room temperature or on ice? There seems to be a major response occurring during the irradiation time although it must take only 1-2 minutes at 80Gy/min. The authors should clarify this point.
4) Figure 2B: the authors state that the response at 2h is similar at the different doses, but this is not the case.
5) Figure 3A is too small and too complex to extract any meaningful information. Since the novel part of this work is the phospho-proteome, perhaps the authors should highlight these results and focus on the more interesting protein families. Figure3B does not seem necessary.
6) The interaction map in Figure 4B is too complicated to extract any meaningful information. How was this constructed? I assume this is the interaction map in the absence of phosphosites. Is that correct? How would you expect this to change after phosphorylation?
7) The discussion needs to be extended to discuss the findings in light of earlier work.
English language is good. A few typos and grammatical errors need to be corrected.
Reviewer 2 Report
This is an outstanding paper in which Shihao Liu and coworkers have shed lights on the proteome and phosphoproteome changes in gamma-irradiated Deinococcus radiodurans, an unusual bacteria that can tolerate extreme stress conditions, including ionizing radiations. The molecular mechanism of its environmental tolerance, however, has remained largely unknown. In this paper, the authors use label-free quantitative proteomics to obtain a detailed view of the proteins and phosphorylation states, as affected by various doses of radiation. They also map the sites of phosphorylation, and perform a thorough statistical analysis of the hits. Although not unexpected, the phosphosites were shown to be present in the regions of intrinsic disorder. Overall, these results have opened new molecular windows on radiation resistance in living cells, which should be useful for future research in this field, particularly in view of the global climate change.
I have a few minor comments, mostly related to presentations, as listed below.
The authors should discuss whether such studies have been done in a standard enteric bacteria, such as E. coli at a low dose radiation.
Line 159-160: This sentence is incomplete, or probably, "enriched" should be "were enriched".
Fig. 4B: The outermost circle is impossible to read due to low resolution and small size. Consider presenting an enlarged version as Additional/ Supplementary file.
In Supplementary Fig. captions (legends):
Fig S1A: Add: Each radiation state is color coded, and the two experiment repeats are shown side by side.
Fig S2 and S3: Describe what the three colors are.
Lastly, the references (Bibliography) are either incomplete or does not follow IJMS format.
Round 2
Reviewer 1 Report
The authors have suitably addressed my remarks in this revised version.